# The Production of *Helianthus* Haploids: A Review of Its Current Status and Future Prospects

**DOI:** 10.3390/plants11212919

**Published:** 2022-10-29

**Authors:** Andrey O. Blinkov, Nataliya V. Varlamova, Ludmila V. Kurenina, Marat R. Khaliluev

**Affiliations:** 1All-Russia Research Institute of Agricultural Biotechnology, Timiryazevskaya 42, 127434 Moscow, Russia; 2Department of Biotechnology, Institute of Agrobiotechnology, Russian State Agrarian University—Moscow Timiryazev Agricultural Academy, Timiryazevskaya 49, 127550 Moscow, Russia

**Keywords:** sunflower (*Helianthus annuus* L.), androgenesis, gynogenesis, embryogenesis, shoot organogenesis, plant growth regulators, γ-irradiated pollen, distant hybridization, ploidy determination

## Abstract

The genus *Helianthus* comprises 52 species and 19 subspecies, with the cultivated sunflower (*Helianthus annuus* L.) representing one of the most important oilseed crops in the world, which is also of value for fodder and technical purposes. Currently, the leading direction in sunflower breeding is to produce highly effective heterosis F_1_ hybrids with increased resistance to biotic and abiotic stresses. The production of inbred parental lines via repeated self-pollination takes 4–8 years, and the creation of a commercial hybrid can take as long as 10 years. However, the use of doubled haploid technology allows for the obtainment of inbred lines in one generation, shortening the time needed for hybrid production. Moreover, it allows for the introgression of the valuable genes present in the wild *Helianthus* species into cultivated sunflowers. Additionally, this technology makes it possible to manipulate the ploidy level, thereby restoring fertility in interspecific hybridization. This review systematizes and analyzes the knowledge available thus far about the production of haploid and dihaploid *Helianthus* plants using male (isolated anther and microspore cultures) and female (unpollinated ovaries and ovules culture) gametophytes, as well as by induced parthenogenesis using γ-irradiated pollen and interspecific hybridization. The genetic, physiological, and physical factors influencing the efficiency of haploid plant production are considered. A special section focuses on the approaches used to double a haploid chromosome set and the direct and indirect methods for determining the ploidy level. The current analyzed data on the successful application of haploid sunflower plants in breeding are summarized.

## 1. Introduction

*Helianthus* is a large genus of the family Asteraceae, comprising 52 species and 19 subspecies [1]. Among them, the cultivated sunflower (*Helianthus annuus* L.), Jerusalem artichoke (*H. tuberosus* L.), and several ornamental species (*H. argophyllus* Torr. & A. Gray, *H. debilis* Nutt., *H. decapetalus* L., *H. maximiliani* Schrad., *H. petiolaris* Nutt., and *H. salicifolius* A. Dietr) are of practical importance [2,3]. The *Helianthus* species of major value is *H. annuus* L., as the most important crop. In fact, sunflower is fourth on the list of cultivated species (accounting for 10%) after palm (38%), soybean (27%), and rapeseed (15%) based on the volume of produced plant oil [1]. According to the Food and Agriculture Organization [4], the global bulk yield of sunflower seeds in 2020 was over 50 million tonnes from an area of 27.8 million hectares. An important byproduct of oil extraction from sunflower seeds is the oil cake, or sunflower meal, widely used as fodder for cattle. Sunflower meal is available worldwide. The estimated global production of sunflower meal in 2019 was 21.85 million tonnes, with Ukraine and the Russian Federation accounting for approximately 7 and 5.1 million tonnes, respectively [5]. In addition, sunflower is grown for direct consumption (confectionery type), as a raw material for use in cosmetics, dyes, greasing substances, and biodiesel, and as an ornamental plant for its cut flowers [3,5]. Currently, sunflower is cultivated worldwide. The main cultivation areas are in the Russian Federation, Ukraine, and Argentina (30.8, 21.8, and 6.9%, respectively) [4].

The basic direction in sunflower breeding is to create highly productive F_1_ heterosis hybrids [6]. The main priorities for the new genotypes are a high yield and seed quality, as well as resistance to herbicides, pests (the European sunflower moth *Homoeosoma nebulellum*, sunflower stem weevil *Smicronyx fulvus*, and others), and diseases, such as downy mildew (*Plasmopara helianthin*), verticillium wilt (*Verticillium dahlia*), rust (*Puccinia helianthin*), Alternaria leaf spot (*Alternaria helianthi*), phoma black stem (*Phoma macdonaldii*), and the parasite broomrape (*Orobanche cumana*) [2,5,7]. In turn, fungal and bacterial pathogens and broomrape develop new virulent races that are able to overcome the resistance genes of available hybrids, thereby significantly decreasing the yield and quality. This situation demands the design of new genotypes, and this makes sunflower breeding a dynamic process [8]. Developing self-pollinated inbred lines for use as the parental components of hybridization and in the production of F_1_ hybrids requires, on average, 4–8 years [9,10], while the creation of a hybrid may take as long as 10–12 years [11]. Currently, long-term self-pollination using either paper or cloth isolators is the most widely applied approach [3]. The development of pure lines can be accelerated by additionally utilizing immature embryo rescue, which allows for up to four generations over a year [12]. The production of homozygous lines using the doubled haploid technique reduces the time interval to 1–2 years [10,13,14,15,16].

The polyploid series of the genus *Helianthus* includes diploids (2*n* = 2*x* = 34), tetraploids (2*n* = 4*x* = 68), and hexaploids (2*n* = 6*x* = 102). Different ploidy levels present certain difficulties in the distant hybridization of sunflower [6]. As for the doubled haploid technique, this approach makes it possible to decrease the ploidy levels of some wild species and interspecific hybrids without a loss of fertility [13,17]. Additionally, it facilitates interspecific hybridization during the introgression of valuable resistance genes from wild species [17]. An efficient protocol developed for the production of sunflower doubled haploids enhances the distant hybridization of species differing in their ploidy level to obtain sound, fertile, interspecific hybrids, as was achieved for potato [18].

Mutagenesis is successfully applied to sunflower to improve several agronomic traits, such as the plant height, qualitative and quantitative oil composition, and resistance to diseases and herbicides [3]. The placement of cultured haploid cells and tissues of agricultural crops on culture media supplemented with mutagens enables the controlled selection of valuable traits determined by both recessive and dominant genes [19]. Correspondingly, a combination of technologies for sunflower haploid production and mutagenesis may represent a powerful and efficient tool for enriching genetic diversity.

Davey and Jan [20], in their review, reported that the US National Sunflower Association (NSA), in 2009, approved an investment of USD 250,000 for the development of an efficient protocol for the production of sunflower doubled haploids. This project was announced on the NSA site in the section on NSA Funded Research (2011–2014). The main goal of this project was to develop an efficient protocol that would be suitable for application in the field by breeding companies [21].

A number of reviews describe the production of sunflower haploid plants [6,22,23,24,25,26,27,28,29,30,31]. However, the information in these reviews is very concise and rather incomplete. Detailed reviews on the production of sunflower haploids were published at least 30 years ago [9,32] and lack many important aspects that have been learned since. In this review, we attempt to systematize and analyze the currently available knowledge in the area of haploid technology related to *Helianthus* plants.

## 2. Production of Sunflower Haploids in Isolated Anther Cultures

Although the first information regarding doubled haploid *Helianthus* plants in relation to the culture of isolated anthers appeared as early as the 1980s [33,34,35,36,37], this technology cannot currently be regarded as efficient. This can be explained by various genetic, physiological, and physical factors that influence the formation of true sunflower haploid plants from microspores.

The doubled haploids of sunflower can be produced in vitro from an anther culture by direct embryogenesis [9,17,38,39], indirect embryogenesis [13,39,40,41,42,43,44,45,46], or indirect shoot organogenesis [10,17,38,40,44,47,48,49,50,51,52] (Figure 1). Several independent research groups have reported the formation of callus tissue from anther culture without any subsequent morphogenic response (Table 1 and Table 2) [9,17,30,41,42,43,44,47,48,49,50,53].

The time interval from the beginning of anther culture to callus tissue formation and the subsequent organogenesis or embryogenesis varies considerably. In particular, callus is formed in vitro on days 5–10, depending on the genotype [40,42,51], on days 10–15 [45,54], on days 15–20 [14,52], and after 4 weeks of cultivation [41,44]. Indirect shoot organogenesis or embryogenesis is observable after 4 weeks of culture [10] and later, after 80–100 days [52] or 120 days [51]. Several researchers have noted that the complete cycle from the beginning of anther culture to the adaptation of in vitro regenerants to the soil conditions is rather long, lasting for up to 16–19 weeks [17].

**Table 1 plants-11-02919-t001:** Callus formation and shoot organogenesis in sunflower anther culture.

Genotypes	Callus Induction Medium (CIM)	Shoot Induction Medium (SIM)	Results	Reference
Cultivar of *H. annuus* L., 2 F_1_ interspecific hybrids, 11 species of *Helianthus*	Modified Murashige and Skoog (MS) medium [55] + White’s vitamins [56] + 1 mg/L 2,4-dichlorophenoxyacetic acid (2,4-D) + 0.2 mg/L kinetin + 30 g/L sucrose	Modified MS + White’s vitamins + 2 mg/L 6-benzylaminopurine (BAP) + 0.2 mg/L indole-3-acetic acid (IAA) + 30 g/L sucrose	Non-morphogenic callus	[9]
Two F_1_ *H. annuus* L. hybrids	B1 (modified MS + White’s vitamins + 5 mg/L zeatin + 30 g/L sucrose)	P20 (modified N6 [57] + 1 mg/L zeatin + 31.65 g/L maltose) or BRl (modified MS + 0.1 mg/L 1-naphthaleneacetic acid (NAA) + 0.1 mg/L BAP + 0.01 mg/L gibberellic acid (GA_3_) + 30 g/L sucrose)	Callus formation in 65–68% of anthers, no shoot organogenesis	[45]
F_1_ hybrid *H. annuus* L.	M1 (½ MS macro- and microsalts + vitamins of Morel and Wetmore [58] + 0.5 mg/L NAA + 0.5 mg/L BAP + 120 g/L sucrose)	P20 or BRl	Callus formation in 95.7% of anthers, 3.4% shoot organogenesis, no data on ploidy level of plants	[45]
Two *H. annuus* L. inbred lines	M3 (½ MS macro- and microsalts + vitamins of Morel and Wetmore + 0.5 mg/L NAA + 0.5 mg/L BAP + 120 g/L sucrose)	P20 or BRl	Callus formation in 20–70%, 1.4–3.3% shoot organogenesis, no data on ploidy level of plants	[45]
One species of *Helianthus*, two interspecific F_1_ hybrids	MS + 0.5 mg/L NAA + 0.5 mg/L BAP	Gamborg (B_5_) medium [59]	Callus formation in 14.7–52.3%, 180 plantlets regenerated and rooted in one hybrid, mostly diploids	[52]
Interspecific hybrid	MS-I1 (MS + 0.5 mg/L BAP + 0.5 mg/L NAA + 30 g/L sucrose)	MS-R1 (MS + 0.5 mg/L BAP + 0.5 mg/L NAA + 30 g/L sucrose)	Callus formation in 96%, 187 regenerated shoots, 44 rooted plantlets, androgenetic origin of examined plants	[38]
Interspecific hybrid	MS-I4 (modified MS + 1 mg/L BAP + 1 mg/L NAA + 30 g/L sucrose)		Non-morphogenic callus in 96%	[38]
Interspecific hybrid	MS-I5 (MS + 0.5 mg/L BAP + 0.5 mg/L NAA + 30 g/L sucrose)	MS-R1	Callus formation in 75%, 85 regenerated shoots, 41 rooted plantlets, androgenetic origin of examined plants	[38]
Six interspecific F_1_ hybrids	L4 [60] + 30 g/L sucrose	B_5_ + 10 g/L sucrose	Callus formation in 50–100% with shoot regeneration in four hybrids, no data on ploidy level of plants	[51]
Cultivar of *H. annuus* L., two interspecific hybrids, three wild species	MS + 1 mg/L NAA + 2 mg/L 2,4-D + 0.5 mg/L BAP + 30 g/L sucrose	MS + 0.5 mg/L BAP + 30 g/L sucrose	Only one interspecific hybrid developed plantlets, no data on ploidy level of plants	[40]
Interspecific F_1_ hybrid	Modified MS + 1 mg/L BAP + 1 mg/L NAA + 30 g/L sucrose	MS + 0.5 mg/L BAP	Callus formation with 1.2% shoot regeneration, haploid plants	[17]
Five fertility restorer lines of *H. annuus* L., five wild species	MS + 1 mg/L NAA + 2 mg/L 2,4-D + 0.5 mg/L BAP + 30 g/L sucrose	MS + 0.5 mg/L BAP + 0.5 mg/L kinetin	Anthers of all genotypes were showed callus, shoot, and root regeneration in only two species, all plants were haploids	[48]
74 cultivated sunflower plants in BC_2_ backcross generation	MS + 0.5 mg/L BAP + 0.5 mg/L NAA + 30 g/L sucrose		Shoot regeneration of 9.39% in 48.65% of the tested genotypes, no data on ploidy level of plants	[10]
Two *H. annuus* L. cultivars	MS + 2 mg/L NAA + 1 mg/L BAP		Callus formation in 9–99%	[53]
Five *H. annuus* L. lines	Modified MS + 1 mg/L IAA + 2 mg/L 2,4-D + 0.5 mg/L BAP + 30 g/L sucrose	MS + 1 mg/L kinetin + 0.1 mg/L IAA	Shoot regeneration of 21.03%, no data on ploidy level of plants	[49]
F_1_ *H. annuus* L. hybrids	MS + 2 mg/L NAA + 1 mg/L BAP		Callus formation in 90%, no shoot organogenesis, callus contained haploid and diploid cells	[14]
Two F_1_ *H. annuus* L. hybrids	MS + 2 mg/L NAA + 1 mg/L BAP + 30 g/L sucrose or MS + 2 mg/L NAA + 0.5 mg/L BAP + 30 g/L sucrose		Callus formation in 34.95%, no data on ploidy level of plants	[50]
Seven *H. annuus* L. F_2_ segregation populations	MS + 2 mg/L NAA + 2 mg/L 2,4-D + 0.5 mg/L BAP + 30 g/L sucrose		Callus formation in 8.3–66.7%, shoot regeneration of 0–6.67%, no data on ploidy level of plants	[50]

**Table 2 plants-11-02919-t002:** Direct and indirect embryogenesis in sunflower anther culture.

Genotypes	Embryogenesis Induction Medium (EIM)	Embryo Development Medium (EDM)	Results	Reference
One *Helianthus* species, one interspecific hybrid	Modified MS medium + White’s vitamins + 5 mg/L zeatin + 30 g/L sucrose		Direct embryogenesis, 1–3 regenerated plants, different chromosome numbers	[9]
Three *H. annuus* L. lines, five F_1_ *H. annuus* L. hybrids	½ MS + vitamins of Morel and Wetmore + 0.5 mg/L NAA + 0.5 mg/L BAP + 120 g/L sucrose	Liquid medium with filter paper: macro- and microsalts of Monnier [61] + vitamins of Morel and Wetmore + 15 g/L sucrose	Embryogenic anthers in 2.1–54.9%, plants had haploid and diploid chromosome numbers	[13]
Two interspecific hybrids	M1 (½ MS macro- and microsalts + vitamins of Morel and Wetmore + 0.5 mg/L NAA + 0.5 mg/L BAP + 120 g/L sucrose)	P20 (modified N6 + 1 mg/L zeatin + 31.65 g/L maltose) or BRl (modified MS + 0.1 mg/L NAA + 0.1 mg/L BAP + 0.01 mg/L GA_3_ + 30 g/L sucrose)	Callus formation in both hybrids, one embryoid formed shoots, no data on ploidy level	[45]
Four *H. annuus* L. genotypes	MS + 1 mg/L 2,4-D + 0.5 mg/L BAP + 40 g/L sucrose	MS + 0.5 mg/L kinetin + 0.5 mg/L BAP	All genotypes produced callus, 10–15% of embryos germinated into plantlets for one genotype, no data on ploidy level of plants	[41]
Seven F_1_ hybrids, four *H. annuus* L. inbred lines	½ MS macrosalts + MS microsalts + vitamins of Morel and Wetmore + 0.5 mg/L NAA + 0.5 mg/L BAP	Monnier medium + 0.05 mg/L BAP + reduced sucrose concentrations (10% for first week, 6% for second, 3% thereafter)	Callogenic anthers in 16–18%, embryogenic anthers in 1–11%, direct and indirect embryogenesis, all plantlets were diploid and originated from somatic cells	[39]
Two interspecific hybrids	MS-I3 (modified MS + 0.5 mg/L BAP + 0.5 mg/L NAA + 30 g/L sucrose)	MS-R3 (modified MS + 30 g/L sucrose)	Direct embryogenesis in up to 92.7%, androgenetic origin of examined plants	[38]
One cultivar, three *H. annuus* L. hybrids	Callus formation: MS + 2 mg/L NAA + 1 mg/L BAP + 30 g/L sucrose; indirect embryo initiation: MS + 0.1 mg/L NAA + 0.5 mg/L BAP	MS + 0.5 mg/L BAP	Calli produced in >90% of anthers, 44% of calli differentiated into embryos, low rate of embryo development, plants were haploid and diploid	[42]
Interspecific F_1_ hybrid	Modified MS + 1 mg/L BAP + 1 mg/L NAA + 30 g/L sucrose	MS + 0.5 mg/L BAP	All anthers developed embryos, regeneration of 98.7%, haploid plants	[17]
Six sunflower cultivars	MS + 0.5 mg/L BAP + 0.5 mg/L NAA + 30 g/L sucrose		Callus formation in 50–90%, embryo formation in 0–51.7%, no data on ploidy level of produced plants	[46]
*H. annuus* L. cultivar	MS + 1 mg/L BAP + 2 mg/L NAA + 30 g/L sucrose		Callus formation in 60.44%, indirect embryogenesis in 20.45%, no data on ploidy level of produced plants	[43]
Two *H. annuus* L. cultivars	MS + 0.5 mg/L 2,4-D + 0.5 mg/L BAP + 30 g/L sucrose	MS medium + 0.5 mg/L kinetin	Embryogenic callus formation in 81–88%, regeneration in 2–9%, no data on ploidy level of plants	[44]

Garkusha et al. [50] characterized the types of calli developed on one hybrid, which were either morphogenic or non-morphogenic. A dense, globular, white-colored callus, a watery, transparent callus with yellowish zones of meristematic cells, and a dense, opaque, green- or red-colored callus were regarded as morphogenic calli. The hyperhydrated friable calli colored yellow-brown or white were deemed non-morphogenic. Hyperhydrated friable calli with dense white meristematic zones were regarded as partially morphogenic. This classification agrees with the data in other studies [41].

The morphogenetic efficiency of an in vitro sunflower anther culture essentially depends on different genetic, physiological, and physical factors, which are described in the next section.

### 2.1. Genotype

Most researchers experimenting with haploid sunflower plants observe a considerable dependence of the anther morphogenic response and haploid plant production on the genotype [10,13,15,45,49,50,51,53]. Saji and Sujatha [42] explained that this is due to the different contents and concentrations of endogenous phytohormones. In general, interspecific sunflower hybrids are more responsive to the induction of androgenesis compared with the F_1_ hybrids of cultivated sunflower [13,15]. Bohorova and Atanassov [9] showed that anther cultures of hybrid sunflower are more efficient in forming callus tissue in terms of both the intensity and the rate of the process compared with sunflower lines and cultivars. The dependence of morphogenetic responses on the specific genetic features of individual sunflower genotypes has been described in several research articles [62,63,64,65]. When studying the genetic control of organogenesis in sunflower, Sarrafi et al. [62] observed that both the general and specific combining ability were significant for shoot regeneration and rhizogenesis. Evidence of a cytoplasmic effect on certain organogenesis parameters and the nucleus–cytoplasm interaction has also been described [63]. Flores Berrios et al. [64] mapped four regions related to in vitro regeneration via organogenesis in recombinant inbred lines. Recently, Petitprez et al. [65] reported on the genetic control of somatic embryogenesis using thin layers of epidermal cells from sunflower hypocotyl as an explant source.

### 2.2. Conditions for Growing Donor Plants

The donor sunflower plants used to produce anthers are grown under both field [9,13,42,44,49] and greenhouse [40,41] conditions. Miladinovic et al. [66] showed that the earlier sowing of sunflower plants under field conditions makes the anthers more responsive. However, the sowing density has no effect on subsequent morphogenesis in anther culture.

### 2.3. Stages of Microspore Development

Currently, there is valid evidence to suggest that the age of the anthers (and, correspondingly, the developmental stage of the contained microspores) plays an important role in the successful production of haploid plants (Figure 2).

Most researchers recommend using immature anthers carrying medium to late uninucleate microspores (i.e., directly before the first mitotic division; Figure 2D) [9,13,51]. Nurhidayah et al. [38] achieved the best results with anthers of distant hybrids that carried early uninucleate microspores (Figure 2C). However, Gürel et al. [45] reported a high rate of callus formation and shoot organogenesis using anthers with the most microspores at the tetrad stage (Figure 2B) compared with anthers containing uninucleated microspores. It should be noted that for the induction of the morphogenesis of some genotypes, it was critically important that the microspores were in the tetrad stage. Several researchers have shown that the stage between the dyad and tetrad is the best for subsequent callus formation and direct embryogenesis (Figure 2A) [13].

Several morphological traits are distinguished when selecting inflorescences and anthers with microspores in certain developmental stages. In particular, a morphological characteristic is the size of the capitulum, which varies from 0.5 to 5 cm depending on the sunflower genotype [40,50]. Another important characteristic is the light yellow [50], cream [40], or light green [13] coloring of the anthers. Several researchers [43,53] have proposed selecting sunflower inflorescences at the R-5.1 stage of development, according to the classification by Schneiter and Miller [67]. The flower size positively correlates with the stage of microspore development (longer florets carry more microspores at the medium to late developmental stages) [43]. Dayan and Arda [14] recommended selecting inflorescences on day 70 after the plants sprout and using anthers isolated from the flower buds with a length of 3–4 mm. Some authors propose collecting the capitula before the ray flowers open [41,42].

Regardless of the proposed optimal stage of microspore development, it should be kept in mind that anther development is asynchronous, not only among different floret bud rows but also within one floret bud or even one anther [9]. In particular, cytological data demonstrate that the number of microspores at the medium to late uninucleate stage in the anthers of the first, second, and third outermost rows of the sunflower head amount to 39.33, 26.0, and 15.33%, respectively. In addition, the anthers of all three rows have been shown to form calli at different rates in vitro (41.0, 39.9, and 28.5%, respectively); however, only the anthers of the first and second rows emerge as morphogenic [43].

### 2.4. Cold Pretreatment of Inflorescences and Flower Buds

An important condition required for microspores to switch from a gametophytic to a sporophytic developmental pathway is the stress pretreatment of the plants [41,42,50] or flower buds [44]. The stress pretreatment of inflorescences or flower buds at a low positive temperature is the most efficient for sunflower. In particular, Çakmak et al. [44] exposed flower buds to 35 °C for 2 days and observed a negative effect on callus formation and shoot regeneration compared with a 24 h pretreatment at +4 °C. In addition, the cold pretreatment of inflorescences stimulated the induction of embryogenesis [41]. The cold pretreatment of inflorescences for 48 h guarantees a maximum number of formed embryos; however, an increase in the duration of exposure has a negative effect on embryogenesis [41]. On the contrary, Saji and Sujatha [42] observed a fourfold increase in the rate of embryogenesis, rising to 14.7 and 21.7% (compared with 7% in the control), after 3- and 4-day exposure to cold temperatures, respectively.

### 2.5. Composition of Culture Medium

The data listed in Table 1 and Table 2 demonstrate that the composition of the culture medium has a crucial effect on the morphogenesis efficiency of the anther cultures of different sunflower genotypes. Independent of the direct or indirect morphogenesis pathway in sunflower anther cultures, two culture media differing in their composition are most frequently used: callus induction medium (CIM), or embryo induction medium (EIM), and shoot induction medium (SIM) for callus tissue [13,17,38,39,40,41,42,44,45,47,48,49,51,52]. The use of anther culture in the same medium for callus induction, shoot regeneration, embryogenesis, and further development (EDM) is relatively rare [9,46].

A major part of both CIM and EIM is the basal components of Murashige and Skoog (MS) [55] culture medium [10,14,40,41,42,43,44,46,47,48,50,52,53]. As has been demonstrated, basal MS medium provides the best results for morphogenesis induction as compared with Gamborg (B_5_) or Nitsch media [41,54]. Note that anther culture on White’s medium failed to induce any callus formation or morphogenetic processes [41]. Some researchers recommend using a modified MS medium. The differences consist of (1) halving the macro- and microsalts [13,39,45], (2) increasing the concentration of potassium ions [49], (3) increasing the concentrations of several vitamins [38], and (4) replacing the vitamins recommended by the MS protocol with the vitamin set of the White [9,45] or Morel and Wetmore [13,39,45] protocol.

The culture media used for the induction of direct embryogenesis or shoot organogenesis are, in many cases, similar in their composition of macro- and microsalts, but they may differ in their composition of vitamins and plant growth regulators (PGRs) [9,13,17,38,40,41,42,44,45,48,49]. The basic components of the N6 [45], B5 [47,51,52], and MR [39] media are considerably less frequently used as basic components for EIM and SIM.

As a rule, the culture media contain a rich set of amino acids. The most frequently observed components are casein hydrolysate at a concentration of 100–500 mg/L [9,40,41,45,48,49,50,53] or individual amino acids, such as L-asparagine, L-glutamine, L-serine, L-tryptophan, or L-cysteine [9,13,39,45]. Coconut water, which contains amino acids, carbohydrates, fats, vitamins, and minerals, has also been successfully used at a concentration of 100 mg/L [43].

The type and concentration of carbohydrates in the culture medium are of considerable importance. Saji and Sujatha [42] and Kostina et al. [49] suggested sucrose as a preferable source of carbon as compared with glucose or maltose. As a rule, anthers are cultured on medium containing 3% sucrose [9,10,17,38,40,42,44,45,48,49,51]. No morphogenetic response was observed in medium containing 2% sucrose [41]. The data on the effect of an increased sucrose concentration on the induction of morphogenesis are contradictory. In some studies, a stimulatory effect of 4% sucrose on the induction of embryogenesis was observed [41,42]. Increasing the sucrose concentration to 80 g/L considerably decreased the rate of embryogenesis in individual sunflower genotypes [41]. On the contrary, Jonard and Mezzarobba [13] reported the successful formation of regenerants from anthers cultured in medium supplemented with 120 g/L sucrose. The successful development of embryos from anther cultures was observed in medium with a reduced sucrose concentration (10% for the first week, 6% for the second week, and 3% thereafter) [39].

Anthers are cultured on solid medium with the pH varying in the range of 5.6–5.9 [13,15,38,40,41,42,45,46,66]. When anthers are cultured in liquid medium, they swell in the early stages, followed by browning and shrinking [41]. Agar (6–7 g/L) [42,45,53], Gelrite (3–3.3 g/L) [10,15,38,46], or Phytagel (2 g/L) [44] can be used as gelling agents. According to several researchers, increasing the concentration of agar in the culture medium to 8 g/L has a positive effect on the rate of callus formation [40,48,52].

Auxin and cytokinin PGRs are necessary components of any culture medium for the induction of morphogenesis. As has been repeatedly demonstrated, callus formation definitely requires exogenous PGRs [14,40]. Note that the selection of particular PGRs and their concentration essentially depends on the genotype (Table 1 and Table 2). An analysis of the published data suggests that the morphogenetic response in sunflower anther culture is most frequently induced on a medium supplemented with 6-benzylaminopurine (BAP) in combination with 1-naphthaleneacetic acid (NAA) at a concentration of 0.5 mg/L [10,17,38,42,46]. When used separately, PGRs fail to induce callus formation [14]. In addition, any change in the concentration of the PGRs does not yield positive results. In particular, increased BAP and NAA concentrations enhance the formation of callus tissue at a higher rate in earlier culture stages, but no shoot organogenesis is observed later [14,38]. Several researchers have shown that increasing the BAP concentration to 2 mg/L has a negative effect on callus formation [50] and decreasing the BAP to 0.1 mg/L reduces the rate of callus formation to 57% compared with medium containing 0.5 mg/L BAP (96%) [13]. Saji and Sujatha [42] transferred embryos at the cotyledonary stage to MS medium supplemented with 5 mg/L BAP for multiple shoot formation. Subsequently, four to six shoots developed from each embryo.

Replacing NAA with auxin PGRs has been shown to be ineffective. In particular, Thengane et al. [41] obtained loose non-morphogenic calli when different concentrations of NAA and 2,4-dichlorophenoxyacetic acid (2,4-D) were added to the medium. Çakmak et al. [44] believed that the negative effect of particularly high PGR concentrations (0.5 mg/L 2,4-D and 2 mg/L NAA) on callus development could be explained by their genotoxic effect, which was confirmed by a DNA comet assay.

The efficiency of other cytokinins in the induction of callus formation and in organogenesis and embryogenesis is ambiguous. In particular, the use of kinetin has no positive effect on the indirect shoot regeneration of different sunflower genotypes [40]. Adding zeatin at a high concentration (5 mg/L) to the induction medium resulted in a high rate of callus formation in two sunflower inbred lines and two hybrids; however, the formed calli were found to be non-morphogenic [45]. On the contrary, Bohorova and Atanassov [9] observed the direct embryogenesis of *H. divaricatus* L. and *H. annuus* L. × *H. decapetalus* L. interspecific hybrids on medium with the same zeatin concentration.

Difficulties in shoot regeneration or the formation of embryos from calli have been repeatedly reported [9,38,45,51]. For example, Saji and Sujatha [42] explained that these may be due to the suppression of the radicular end, causing the hyperhydration of the cotyledons and/or callusing of the embryos. Gürel et al. [45] assumed that the low efficiency of sunflower haploid production was associated with a high rate of ethylene biosynthesis during anther culture. According to Vasić et al. [46], using 1 mg/L silver nitrate as an inhibitor of ethylene biosynthesis is preferable to using 0.1% polyvinylpyrrolidone (PVP). Several papers recommend increasing the AgNO_3_ concentration to 2–10 mg/L [50] or 10 mg/L [49] depending on the genotype.

Various phenolic compounds accumulate during anther culture, and this leads to the browning of the tissues and culture medium due to their oxidation by polyphenol oxidases and peroxidases. The swelling of the anthers (during the first week), followed by browning, has been described [14,39,42]. However, the browning does not interfere with callus formation [42]. Adding 0.1% PVP to the medium reduced the browning of the anther and enhanced embryogenesis [39].

An important step in the production of haploid plants in sunflower anther cultures is the rooting of the regenerants, followed by clonal propagation and adaptation to the soil conditions. Todorova et al. [51] described the difficulties associated with the rooting of haploid plants. Some researchers reported the death of rooted *H. mollis* Lam. regenerants after transferring them to soil [52]. Sujatha and Prabakaran [17] recommended using ½ MS medium supplemented with 1 mg/L NAA for the successful root induction of regenerants from *H. resinosus* Small × *H. annuus* L. and *H. tuberosus* L. *× H. annuus* L. interspecific hybrids.

Bohorova and Atanassov [9] proposed solving the difficulties associated with the clonal micropropagation of produced haploid plants via secondary callus induction and shoot organogenesis from the haploid regenerants. The shoot fragments of haploid regenerants produced by *H. annuus* L. (2*n* = 34) × *H. divaricatus* L. (2*n* = 68) anthers were cultured on medium containing macro- and microsalt MS with modified NaFe-EDTA (5 mL/L solution from stock containing 5.57 g/L FeSO_4_ × 2H_2_O and 7.75 g/L Na_2_EDTA), the vitamins according to White, 500 mg/L casein hydrolysate, 500 mg/L myo-inositol, 2 mg/L BAP, 0.2 mg/L indole-3-acetic acid (IAA), 20 mg/L adenine, 1320 mg/L L-asparagine (or 800 mg/L L-asparagine + 800 mg/L L-glutamine), and 30 g/L sucrose. This culture medium composition guarantees multiple shoot regeneration after three passages, followed by successful rooting. On the other hand, a cytological analysis of the root tips of 125 plants produced during the first 15 sub-cultivations demonstrated that the plants resulting from the first 3 sub-cultivations carried 34 chromosomes. Among the plants resulting from passages three to eight, some individuals had 68 and 102 chromosomes. Starting from passage nine, all of them produced regenerants carrying 51 chromosomes, similar to the donor plant. Only one plant was produced and rooted using this protocol in the case of *H. divaricatus* L. [9]. Thus, the successful application of this methodological approach to the induction of secondary callus formation and shoot regeneration in haploid plants depends on the genotype.

### 2.6. Culture Conditions

The temperature, light pattern, and photoperiod significantly influence the morphogenetic response of sunflower anther culture. Note that the genotype plays an important role. Nurhidayah et al. [38] studied the effects of the temperature (30 or 35 °C) and number of days of culture in the dark (0, 6, 12, and 18 days) and suggested that the efficiency of morphogenesis and the optimal culture conditions vary depending on the sunflower genotype.

Todorova et al. [51] and Priya et al. [40] obtained the best results at 25 °C with a 16/8 h (day/night) photoperiod. On the contrary, Saji and Sujatha [42] recommended culturing at 25 °C for 10 days in the dark until the time of callus formation. In some cases, the best result was attained at 30 °C in the dark [41,46,54]. At a higher temperature (35 °C), anther cultures form considerably dehydrated calli with a low regeneration ability [17]. On the contrary, several studies have shown the highest rate of embryogenesis at 35 °C in the dark during the first 12 days [13,39]. Saensee et al. [43] cultured anthers at 25 °C for 5 days in the dark, followed by a 16/8 h photoperiod. However, several studies have shown that incubation in the dark has no considerable effect on indirect organogenesis and embryogenesis [42,53]. In particular, the anthers of three distant sunflower hybrids developed calli in all media when cultured in either the light or dark; however, the calli formed in the dark were non-morphogenic [38]. Note that the color of the callus tissues depends on the light conditions during culture. In particular, white or yellow calli are mainly formed in the dark, and green ones are mostly formed in the light [45].

## 3. Producing Sunflower Haploids in Isolated Microspore Culture

Isolated microspore culture may be the most promising technique for producing doubled haploids, since it is rather simple and cost-efficient. Isolated microspores allow for an abundant yield of independent haploid embryos and, additionally, doubled haploids if a highly efficient protocol is designed for a particular genotype. The development of a true embryoid makes microspore culture an ideal model for studies of in vitro embryogenesis in regard to different basic and applied aspects of plant reproductive biology. In addition, the absence of somatic tissues in in vitro microspore cultures guarantees the haploid nature of the plants [68]. However, studies on the production of haploids via isolated microspore culture are scarce and date back 25 years [69,70,71]. Moreover, no full-fledged haploid plants were obtained using this method. This can be explained by the low viability of microspores at the early development stage of in vitro tissue culture. In particular, Gürel et al. [69] observed an increase in the volume of microspores during 2 days of culture and additional cell wall degradation. A few symmetric and asymmetric divisions were observed, and the cells died within 3–4 days. The development of embryos that formed over 15 days was arrested, and they necrotized and died during the transfer to SIM [69]. Several studies have shown that the viability of microspores during culture depends on the genotype [69,70].

Todorova et al. [70] described callus formation from microspores. Coumans et al. [71] explained the process of callus formation in relation to the multicellular trichomes in the apical part of an anther, which are filtered together with the microspores and are highly responsive to morphogenesis. These hair-type structures of sunflower anthers visually resemble embryoids; the multicellular head appears similar to a pseudo-embryo, and the tube-like extension appears similar to a pseudo-suspensor (Figure 3). The formation of the pseudo-embryogenic structures of trichomes in isolated microspore cultures has been also observed in other crops [72].

Coumans et al. [71] recommended selecting floret buds carrying microspores developed from the tetrad stage to mid-uninucleate vacuolated stage in order to achieve the highest yield of viable microspores. These cells can be isolated beforehand and stored at 4 °C for 14 days without any loss of viability. The use of an osmotic agent in the culture medium when isolating microspores, in particular 12% sucrose, is required [70,71]. In particular, according to Coumans et al. [71], 0.38 M sucrose performs better as an osmotic in the isolation medium than mannitol.

The important physiological factors affecting the success of this technology are the composition of the culture medium and the culture conditions. The best results for sunflower microspore cultures were observed when using basal MS medium with a characteristic high nitrogen content [70]; N6 medium with a high potassium content [71]; and NLN medium [69]. In particular, MS medium supplemented with 800 mg/L glutamine, 100 mg/L serine, and 12% sucrose (pH 6.2) led to a considerable increase in the size of the microspores and their subsequent symmetric division [70]. Culture on N6 medium supplemented with 1 mg/L NAA, 0.2 mg/L BAP, and 0.44 M maltose guarantees the highest rate of viable isolated microspores [71]. The optimal density of the microspores varied from 6 × 10^4^ [70] to 8 × 10^4^ microspores/mL [71].

A likely cause of the absence of morphogenesis in isolated microspore cultures and the death of cultures is the toxic substances released into the culture medium. Cell debris, microspores formed at the tetrad stage and later stages, and mature pollen are possible sources of toxic substances [69]. The darkening of the culture medium during the homogenization of sunflower floret buds has been observed [69,71]. Enzymatic browning is a biochemical process consisting of the oxidation of phenolic compounds to colored quinones by polyphenol oxidases and peroxidases, which are toxic to microspores. These quinones interact with one another as well as with amino acids, proteins, and sugars to produce stable insoluble melanins (brown, black, or red pigments). Sunflower is rich in diverse phenolic compounds, predominantly chlorogenic acid [73]. A decrease in the degree of oxidation of the phenolic compounds is attainable by modifying the culture medium with the addition of antioxidants, which interfere with the oxidation of the phenolic components, or adsorbents, which bind the phenolic compounds and reduce their toxicity [74]. The amount of cell debris and accumulation of toxic substances in the medium can be decreased in several ways, including: (1) being more careful in the selection of floret buds carrying microspores at the early developmental stage (uninucleate vacuolated microspores) [69,71]; (2) using a low density of microspores [69]; (3) changing the isolation medium several times [69] when harvesting the microspores; (4) separating the microspores from the cell debris in Percoll density gradient [71]; or (5) decreasing the oxidation of the phenolic compounds to quinones by using cooled isolation medium, provided that the number of microspores in the medium is no more than approximately one floret bud per 1 mL [71], or by adding an adsorbent to the medium, such as carbon [70]. The use of antioxidants, such as PVP, reduced glutathione, or cysteine, was shown to reduce the viability of microspores [71]. The efficiency of ascorbic acid still remains a questionable issue. The relevant data suggest both positive [70] and negative [71] effects of ascorbic acid on the viability of microspores.

As has been demonstrated, supplementing the medium with Ethrel PGR (50 or 100 µM), which activates ethylene biosynthesis, or with its precursor, aminocyclopropane carboxylic acid (ACC; 100 µM), increases the viability and number of symmetric divisions of microspores. On the contrary, aminoxyacetic acid, which inhibits ethylene biosynthesis, has a negative effect on the division and viability of microspores [71].

## 4. Producing Sunflower Haploids by the In Vitro Culture of Unpollinated Ovaries and Ovules

The foundation for this technology is the in vitro culture of unfertilized sunflower ovules, which results in the haploid embryonic sac cells switching from a gametophytic to a sporophytic pathway, eventually producing embryos (direct embryogenesis) [32,75,76,77,78] or morphogenic calli, followed by embryo or shoot regeneration (indirect embryogenesis or organogenesis) [37]. Although degree of success has been achieved in the production of sunflower haploids using the culture of unpollinated ovaries and ovules, this technology has evidently received insufficient attention, and the available experimental results date back to the 1980s [37,75,76,77,78,79,80,81]. These data are partly consolidated in a review by Yang et al. [32].

Young florets [75], unpollinated ovules [32,75,77,78], and ovaries [37] are used as explant sources for the production of gynogenic haploid plants, and the ovaries are precultured to further isolate ovules [76]. At 1–3 days prior to anthesis, ovaries are used as the donor plant material [76,81]. Experiments show that ovules at this developmental stage carry almost mature morphogenic ovules. According to cytological analysis, these are considerably polarized cells, carrying a nucleus, cytoplasm (accumulated in the chalazal end), and numerous small vacuoles near the micropyle [81].

The efficiency of unpollinated ovules and ovaries depends on the genotype [76]. An important factor in the formation of gynogenic haploid plants is the application of a cold stress pretreatment of the donor inflorescences (12–24 h at 4 °C), which considerably increases the rate of embryoid development [32,77,78].

As in the case of androgenesis, the composition of the culture medium is of paramount importance. N6 [32,75,77,78], MS [37], and MS medium, with minor modifications [76], are typically used as the basal media. The optimal source of carbohydrates for the formation of gynogenic embryos is 12% sucrose [32,77,78].

The absence of exogenous PGRs in the medium is preferable for producing gynogenic haploids from ovules in the case of direct embryogenesis. Note that this concurrently inhibits the development of embryos from somatic cells [32,77,78]. The presence of 2-methyl-4-chlorophenoxyacetic acid (MCPA), an auxin-type PGR, at a concentration of 2 mg/L has been shown to induce the production of exclusively sporophytic embryos [32,77]. Cai and Zhou [75] showed that medium containing kinetin as a cytokinin induced the formation of morphogenic calli from ovule or ovary somatic tissues, which further produced diploid regenerant plants.

In the case of indirect embryogenesis, auxin-type (2,4-D, IAA, and/or NAA) and cytokinin-type (kinetin) PGRs are necessary [37]. Additionally, several researchers recommend supplementing the medium with gibberellic acid (GA_3_) at a concentration of 0.1 or 1.0 mg/L [32,37] for the induction of morphogenesis, as well as coconut milk, which is rich in fats and various vitamins [37]. Table 3 briefly characterizes the available protocols for the production of gynogenic haploids following a direct or indirect embryogenesis pathway.

Gelebart and San [76] described spontaneous chromosome doubling in gynogenic sunflower plants. The rate of haploid cell formation rapidly decreased during the growth of haploid plants obtained from ovules, so that all the cells had become diploid by the time of flowering. Cai and Zhou [75] examined the chromosomes in the root tips and observed the formation of both haploid and diploid plants.

The cytological analysis of unfertilized ovules cultured in vitro showed considerable morphological changes as early as day 5 of cultivation. Note that the development of most cells (approximately 80%) was arrested, followed by degeneration. Viable egg cells that are capable of parthenogenesis are referred to as activated [81]. After 5 days of culture, activated egg cells develop a two- or four-cell proembryo, while the synergids degenerate. On day 7, a stretched suspensor is observable, while the embryoid may occupy the entire space of the embryonic sac by days 8–10. The root is formed and its cells differentiate after 30 days of cultivation [32,82].

Note that the gynogenic and zygotic embryos of sunflower display both similarities and differences in their structure and development (Appendix A). In addition to a comparative study of the development of parthenogenetic and zygotic sunflower embryos [81], the cytological and embryological aspects of the in vitro development of beet (*Beta vulgaris* L.) [83], onion (*Allium cepa* L.) [84], cucumber (*Cucumis sativus* L.) [85], cassava (*Manihot esculenta* Crantz) [86], and some other crop species have also been examined. The specific features of embryo formation from microspores cultures have also been studied, representing a more convenient model for studying gametic embryogenesis compared with the zygotic model [87,88,89].

Gynogenic embryos develop from the side of the micropyle, which confirms the origin of the unfertilized ovules. Sporophytic (diploid) plants can originate from the endothelium, especially in the chalazal part of the ovule, and form callus-like structures. These pseudo-embryogenic structures usually develop after differentiation begins, but their growth is considerably faster. Pseudo-embryogenic structures formed of endothelial cells are very similar to gynogenic embryos in their shape; thus, they are visually indistinguishable when working with unpollinated ovule cultures. Additionally, gynogenic embryos and pseudo-embryogenic structures are morphologically similar [32,77,82].

## 5. Induced Parthenogenesis by γ-Irradiated Pollen

Pollen irradiation is the most efficient technique used to induce haploid sunflower plants in situ. This technique involves the pollination of female florets with radiation-exposed pollen, followed by the isolation of embryos formed from the ovules via parthenogenesis and their in vitro culture. Most frequently, ^137^Cs and ^60^Co are used as the sources of γ-radiation. The production of haploid sunflower plants by γ-induced parthenogenesis was first attempted at the Dobrudzha Agricultural Institute (Bulgaria) in 1993 [90]. In 1997, Bulgarian researchers published the first communication about the successful production of *H. annuus* L. using this technique [91].

The technology for producing doubled haploids using γ-induced parthenogenesis comprises several steps (Figure 4). Initially, the florets of the sunflower capitulum are emasculated either manually [51,92] or chemically [16,91] using a double treatment of the upper pair of leaves and by developing floret buds with a diameter of 1–1.45 cm using GA_3_ (45 mg/L) at an interval of 2 days. A necessary condition is the strict isolation of the emasculated plants to prevent any foreign pollen from affecting the process. Fertility restorer lines are used as the pollen donors [51,91]. The pollen of paternal plants is harvested beforehand and stored at 4 °C until further use. One day before pollination, the pollen is exposed to γ-radiation [93] using either ^137^Cs at a dose of 300, 600, 700, or 100 Gy [51,91,94] or ^60^Co at a dose of 750 or 1000 Gy [92], with a source position of 3.38 Gy/min [95]. Only plants that do not develop their own pollen are pollinated [95]. Embryos formed over 12–24 days after pollination are plated onto a modified MS culture medium without any PGRs (½ macro- and microsalts, vitamin B_5_, and 20 g/L sucrose) [51,91] under a 16/8 h photoperiod and a temperature of 25 °C [51,91,92]. All fertile doubled haploids are self-pollinated [16].

One of the most important factors determining a high efficiency in the production of parthenogenetic plants is an adequate dose of γ-irradiation. The dose should not be lethal, which would completely inhibit pollen tube growth, but high enough to interfere with normal fertilization and prevent the formation of diploid hybrid embryos. It is believed that pollination with radiation-exposed pollen partially induces the cell cycle, thereby inciting DNA replication and the duplication of chromosomes in the absence of cell division. As a result, homozygous diploid ovules develop into parthenogenetic diploid embryoids [96], while the produced plants are fertile doubled haploids [97]. Note that the optimal dose of γ-radiation strongly depends on the specific genetic features of the pollinator plant [51,91,92,93]. In particular, Todorova and Ivanov [93] showed that the γ-irradiation of a mixture of pollen grains from different sunflower genotypes provided the best efficiency of the haploid plants. Todorova et al. [98] confirmed that the haploid plant yield is dependent on the genotype and the dose of γ-radiation. The authors compared the in vitro germination of pollen grains belonging to six doubled haploid lines of fertility restorers before and after γ-irradiation (^137^Cs at doses of 600 and 900 Gy). As they observed, both doses of γ-radiation slowed the pollen tube growth, and the 900 Gy dose had a stronger inhibitory effect. In that study, lines with both a high and low degree of pollen resistance to γ-irradiation, according to the in vitro growth of the pollen tubes, were produced.

The efficiency of parthenogenetic plant development depends on the specific features of not only the paternal but also the maternal genotype [51,91,92,95,96]. Drumeva and Yankov [96] examined lines and hybrids that differed in their types of CMS and postulated that the parthenogenetic response was determined not only by genotype-specific characteristics but also by the specificity of the nuclear–cytoplasmic interaction. In addition, they demonstrated that the frequency of parthenogenetic embryo production was considerably higher when using the maternal form compared with parental lines of hybrids.

An important step in the production of haploid plants using γ-irradiated pollen is the timely isolation of the embryos and their culture on the appropriate culture medium. In this process, part of the embryo dies, with a corresponding decrease in the rate of parthenogenesis in the plants. In particular, Drumeva and Yankov [96] succeeded in obtaining only 46 plants from 71 embryos cultivated in vitro.

One of the important shortcomings of haploid production via γ-irradiation-induced parthenogenesis is the difficulty in proving the origin of the embryos, which can be formed from either the parthenogenetic development of unfertilized ovules or double fertilization. Molecular and isozyme markers can be useful for confirming the parthenogenetic origin of the plants [91,94].

## 6. Induced Parthenogenesis by Distant Hybridization

The most common method used to produce haploid plants is distant hybridization followed by the partial or complete chromosome elimination of one parental species (predominantly in pollinator plants) [99,100]. Correspondingly, the developing embryos carry only a maternal haploid chromosome set. The cellular mechanisms underlying chromosome elimination are still vague [101]. Several hypotheses have been proposed to explain this phenomenon, including asynchrony in the mitotic cycles of the crossed species [102], the degradation of foreign chromosomes by specific host nucleases [103], asynchrony in the synthesis of nuclear proteins, leading to the loss of individual chromosomes [104], the spatial separation of genomes in the interphase [105], species-specific chromosome inactivation [106,107], and the incorrect chromosome assembly of one parent during the metaphase followed by nondisjunction or delay in the anaphase, leading to micronucleus formation in the early stages of embryonic development [108,109]. Surikov and Dunaeva [110] believe that chromosome elimination is a manifestation of post-gamic incompatibility and guarantees the reproductive isolation of the species. It has also been shown that a centromere-specific histone, CENH3 (variant H3), plays a key role in selective chromosome elimination [111,112]. The study of this histone has created an opportunity to produce doubled haploids through its modification. In 2010, this method was demonstrated in a model object, *Arabidopsis thaliana* L., for the first time, but it has not yet been actively used in breeding programs. However, this method holds promise for the future [113].

The production of doubled haploids via the selective elimination of chromosomes during distant hybridization is widely applied to cereals, particularly barley [114], wheat [115], and triticale [116]. Currently, no valid data are available on the production of haploid sunflower plants using this method. Jan et al. [117] suggest that it is practical to use safflower (*Carthamus tinctorius* L.) or Jerusalem artichoke (*H. tuberosus* L.) as a pollinator plant to produce *H. annuus* L. haploids. The fact that sunflower itself is used as a paternal form to induce the parthenogenetic formation of embryos from reduced safflower and lettuce (*Lactuca sativa* L.) ovules favors this hypothesis. In particular, the inoculation of safflower onto sunflower plants followed by pollination with sunflower pollen yielded three seeds, one of which was a normal diploid [118]. It was assumed that the diploid safflower plant was formed as a result of induced ovule diploidization. Several studies report on the efficient use of pollen from different *Helianthus* species to produce haploid lettuce plants [119,120]. In particular, of 25 examined *Helianthus* species, pollination with *H. annuus* L. and *H. tuberosus* L. resulted in the highest frequency of the production of haploid lettuce plants (16 and 19%, respectively). The embryo rescue technique has enabled the production of morphogenic callus tissue from haploid embryos formed by dividing unfertilized ovules on modified MS medium supplemented with 1 mg/L 2,4-D and 1 mg/L NAA or 2 mg/L BAP and 1 mg/L IAA. The haploid status of 23 obtained lettuce regenerants was confirmed by the genome size using flow cytometry, counting the chromosome in the root tips, and the stomata cell size, as well as disturbances in pollen formation.

A possible way of resolving the difficulties in producing pure lines is to develop a protocol for obtaining sunflower doubled haploids through crosses with haplo-inducer lines, as has been applied in maize [121].

## 7. Methods for Doubling the Sunflower Haploid Chromosome Set and Determining the Ploidy Level

An important step in the production of true haploid sunflower plants is the treatment of the shoots or meristems with antimitotics or mitotic inhibitors, chemical substances that interfere with the spindle function and, correspondingly, chromosome segregation to the poles. The most commonly used chemical compound that induces the formation of doubled haploids is colchicine, an alkaloid that binds to β-tubulin, the major protein of the microtubules [122]. The efficiency of haploid chromosome doubling with colchicine treatment essentially depends on the concentration and exposure time. The concentration of colchicine is experimentally selected to ensure that the amount entering the dividing cells prevents chromosome segregation. If the concentration is too low, either the effect will not manifest or it will yield aneuploid cells. On the contrary, a too-high colchicine concentration will cause plant cell death because of its high general toxicity [123].

A number of research papers suggest the optimal conditions for the polyploidization of sunflower plants. For this purpose, researchers applied treatments to either 2- to 5-day-old [124] or 10-day-old [125] diploid sunflower shoots and their triple exposure to 0.4, 0.5, and 0.6% colchicine water solution for 12 h at intervals of 12 h [124]. However, Vardar et al. [125] observed a toxic effect of this colchicine concentration and exposure time and recommended a lower colchicine concentration (0.1–0.3%) and shorter time (5 or 8 h). Autotetraploids were efficiently produced by treating the axillary buds with 0.35% colchicine water solution for 8 h over 3 days [126].

Regardless of the method used, creating sunflower doubled haploid lines implies the necessary confirmation of the cell ploidy used as their source. In particular, embryos or callus tissue developing in anther or ovule cultures can originate from either gametophytic or sporophytic tissues. The latter are not of practical interest with regard to pure lines, since they represent the genetic properties of the initial parent.

There are many direct and indirect methods for determining the ploidy level of the regenerated plants, which are listed in Table 4. Among the former, cytological methods are used to directly count the chromosome number in the cell metaphase plates of the calli, young leaves, or root apical meristems [9,14,17,39,49,66]. Alternatively, the number of DNA in cells is determined using the corresponding equipment [17,39,52,93,96]. Among the latter, different characteristics of the cells, tissues, and whole plants are utilized that closely correlate with the ploidy level of the plant, including differences in the morphological characteristics between donor plants and plants produced in in vitro culture. Plants regenerated from gametes usually show differences between one another and from the donor plant in several phenotypic characteristics. In particular, Bohorova and Atanassov [9] and Nurhidayah et al. [38] distinguished plants regenerated from an anther culture according to traits such as the height, leaf blade length and width, leaf dentation pattern, and the presence of an anthocyanin color. Histological examination aimed at determining the origin of the calli and embryos is also an indirect method for determining the ploidy level. For example, Zhong et al. [39] demonstrated the formation of calli and embryos in anther walls, which gave rise to sporophytic diploid plants. In a study on the production of sunflower doubled haploids via γ-induced parthenogenesis, control emasculated plants that were not pollinated with irradiated pollen provided indirect evidence for the development of unfertilized ovules, since the control plants did not form seeds, as was also the case for the plants with underdeveloped seeds that were fertilized by γ-irradiated pollen without further embryo rescue [93,95].

Of special importance is the use of isozyme and molecular markers for γ-induced parthenogenesis when the chromosomes are doubled as early as the ovule stage. Screening by esterase and 6-phosphogluconate dehydrogenase enzyme markers is performed to confirm the homozygosity of the produced sunflower doubled haploids. The isozyme patterns of these enzymes demonstrated that the analyzed doubled haploids were homozygous for these enzymes, whereas the maternal plant was heterozygous [91]. Nurhidayah et al. [38] found that the menadione reductase isozyme system provided the most information and confirmed the androgenic origin of only 15 of the 1211 regenerants produced. SSR markers allowed for the confirmation of a parthenogenetic or zygotic origin of the plants obtained via γ-induced parthenogenesis. In particular, Drumeva et al. [94], using the primer pairs for the codominant loci SSL 26 and SSL 46, demonstrated clear allelic differences between the parental lines of the Albena hybrid (2607 A × 147 R), the paternal line (937 R) used for pollen irradiation (^137^Cs, 700 Gy), and the produced doubled haploids. The allele specific to the pollen source was not observed in the examined doubled haploid lines. The electrophoretic patterns of these microsatellite markers showed that the produced doubled haploid lines carried specific alleles of the parental hybrid lines but lacked any alleles of the pollen donor. Concurrently, the SSR assay demonstrated the homozygosity of these loci. This confirms that the genetic material of the pollen donor is not involved in embryo formation.

Both spontaneous chromosome doubling in anther or ovule cultures [47,76] and the production of true haploid plants [13,17,48,75] have been observed in sunflower. The production of diploid plants of unknown origins was reported in several investigations [13,17,75]. The fact that the number of haploid cells rapidly decreases during the ontogenesis of haploid plants produced in either anther [42] or ovule [76] cultures, meaning that almost all the cells become diploid by the flowering time, may be regarded as evidence of spontaneous chemical doubling. Saji and Sujatha [42] and Dayan and Arda [14] described the presence of both haploid and diploid cells in calli produced from anther culture, which also indirectly proves spontaneous chromosome doubling in *Helianthus* plants.

Several studies reported on the formation of aneuploids from the anther culture of sunflower interspecific hybrids [13,42]. Jonard and Mezzaroba [13] described plants produced with sets of 51, 55, 60, 64, 65, 66, 67, 69, 70, 73, and 85 chromosomes. Aneuploids have been obtained from anther and microspore cultures of other crops, such as triticale [128], barley [129], wheat [130], and Brassicaceae species [131,132,133]. The mechanism underlying the formation of aneuploids in in vitro culture is rather vague. Frequently, this results from either an abnormal chromosome number in the initial cells (microspores) or deviations in cell division caused by stress cultivation conditions (PGRs and other substances contained in the culture medium) [128].

Although many neoplasms have been produced from sunflower anther cultures, few studies report the experimental data on the ploidy level of the resulting plants, their morphological characterization during individual development, and the seed progeny [10,45,46,50,51].

## 8. Achievements in Sunflower Breeding Programs by the Implementation of Haploid and Doubled Haploid Plants

Although the rate of sunflower haploid production remains extremely low when using particular methods, some success in the application of certain genotypes developed through breeding is evident. In particular, long-term work on the creation of pure lines of sunflower fertility restorers for heterosis breeding was carried out at the Dobrudzha Agricultural Institute (Bulgaria) [28,134]. Producing sunflower doubled haploids via induced parthenogenesis using γ-irradiated pollen has become a routine procedure [90]. Field studies of the progeny of doubled haploids have allowed for the isolation of lines with absolute or increased resistance to downy mildew (races 330, 700, and 731), phomopsis leaf and stem blight, phoma black stem, and the Alternaria leaf spot of sunflower, as well as broomrape races A, B, C, D, and E. Additionally, lines with increased productivity have been bred. Of special value are the sunflower doubled haploid lines with a combination of agronomic characteristics that have been used as parental lines to explore the heterosis of hybrids [8,11,16]. Sunflower pure lines with enhanced imidazolinone tolerance have also been produced [97]. In fact, in 2005–2017, bred fertility restorer lines were used as the initial parental forms for the production of the commercial hybrids Dobrozvet, Biotzvet, Valin, Mihaela, Alpin [90], Sevar [135], and Linzi [136].

Haploid production may play an important role in fertility restoration in distant hybridization. In particular, the interspecific F_1_ hybrid *H. annuus* L. × *H. resinosus* Small (2*n* = 4*x* = 68) had a low fertility (10–20%) [13]. However, another characteristic of haploid plants produced from the anther culture of this F_1_ hybrid is their high fertility (67 and 40% of fertile pollen grains in plants with 34 and 68 chromosomes, respectively) [13].

The aneuploids produced from the anther culture of the interspecific F_1_ hybrid *H. annuus* L. × *H. resinosus* Small (2*n* = 4*x* = 68) carried different numbers of chromosomes (51, 55, 60, 64, 66, 67, 69, 70, 74, and 85). The pollen fertility rate of aneuploid forms with 60, 66, 67, 69, 70, and 74 chromosomes varied in the range of 78–82%. On the contrary, the fertility rate of pollen grains in aneuploids with 51, 55, 64, and 85 chromosomes was 0–12%. Thus, aneuploids offer the opportunity to obtain unique genetic material that can be utilized in breeding [13].

The *Alternaria helianthi* leaf spot of sunflower is a widespread disease. Wild perennial species, such as *H. mollis* Lam., *H. maximiliani* Schrad., *H. divaricatus* L., *H. occidentalis* Riddell, and *H. decapetalus* L., carry the genes conferring resistance to this disease [54]. Selection for resistance to Alternaria leaf spot is limited because of the strong incompatibility of cultivated sunflower and wild species with different ploidy levels and the sterility of the produced hybrids [54]. The introgression of the genes responsible for resistance to this disease from other *Helianthus* hexaploid species (*H. tuberosus* L. and *H. resinosus* Small; 2*n* = 6*x* = 102) into the cultivated diploid sunflower cultivar Morden was performed [17,54]. The cross *H. tuberosus* L. (Acc. no. TUB 03; PI 451980) × *H. annuus* L. cv. Morden and *H. annuus* L. cv. Morden × *H. resinosus* Small (Acc. no. RES 09; PI 468879) produced fertile tetraploid interspecific hybrids (2*n* = (3 + 1)*x* = 68 and 2*n* = (1 + 3)*x* = 68, respectively), while back-crossing them produced the completely sterile triploid BC_1_ progeny. To avoid sterility in the produced BC_1_ hybrids, haploid plants were obtained using the anther culture of interspecific *H. tuberosus* L. × *H. annuus* L. and *H. annuus* L. × *H. resinosus* Small hybrids. The produced haploid plants were diploid and formed normal tetrads, despite a low rate of pollen fertility (Figure 5).

The artificial infection of plants with *Alternaria helianthi* has shown that 68.5 and 24.3% of the plants produced through anther culture from the interspecific *H. tuberosus* L. and *H. resinosus* Small hybrids, respectively, display increased resistance to this pathogen. Note that the resistance to *Alternaria helianthi* in wild sunflower species is of a polygenic nature. Since the tetraploid interspecific hybrid carries *n* chromosomes of the diploid cultivated sunflower and 3*n* chromosomes of the wild hexaploid, it is able to form two types of gametes: one that carries one genome of each parent (*n* = 1*x* + 1*x*) and one with the two genomes of the wild sunflower (*n* = 2*x*). Thus, the produced haploid plants can be also divided into two types according to the carried genomes. Sujatha and Prabakaran [17] assumed that haploid plants produced with an increased number of sterile pollen grains carried the genomes of two different species and, correspondingly, the pollen sterility is associated with meiotic abnormalities in the pollen grains, while haploid plants with a high pollen fertility carry the genomes of exclusively wild hexaploid species. However, both types of sunflower haploid plants are valuable as promising donor breeding materials.

## 9. Conclusions and Future Prospects

In this review article, we summarized and analyzed the experimental data on haploid induction and dihaploid *Helianthus* plant production by various in vitro and in vivo methods. Based on the above discussion, the following central conclusions can be drawn:(1)Inducing parthenogenesis by γ-irradiated pollen is the most efficient method for producing haploid sunflower plants. The dihaploid lines obtained by this method have been involved in commercial breeding programs to produce high-yielding F_1_ hybrids with an enhanced resistance to abiotic and/or biotic stresses. However, several parameters (the dose of γ-irradiation, dependence on the genotype, embryo survival) mean that this method is substantially limited compared to the classical production of inbred parental lines through cycles of self-pollination.(2)Conventional in vitro methods for haploid induction in *Helianthus* plants using male (isolated microspore and anther cultures) or female (unpollinated ovaries and ovules culture) gametophytes are still not widely used in biotechnological practice due to their low embryogenic response. A radical change in this situation could be achieved by identifying genotypes with a strong ability to undergo gametic embryogenesis using a fine-mapping approach with quantitative trait loci and improving them through various genetic engineering strategies, as well as optimizing physiological factors such as the culture conditions of mother plants, the stage of development of the gametic cells, the culture media composition, and the culture conditions, etc.(3)Alternative in vivo methods for haploid induction in *Helianthus* plants via induced parthenogenesis by distant hybridization, including manipulations with the centromere-specific histone (CENH3), which plays a crucial role in uniparental genome elimination during early embryogenesis, will be of great fundamental and practical value in the future.

## Figures and Tables

**Figure 1 plants-11-02919-f001:**
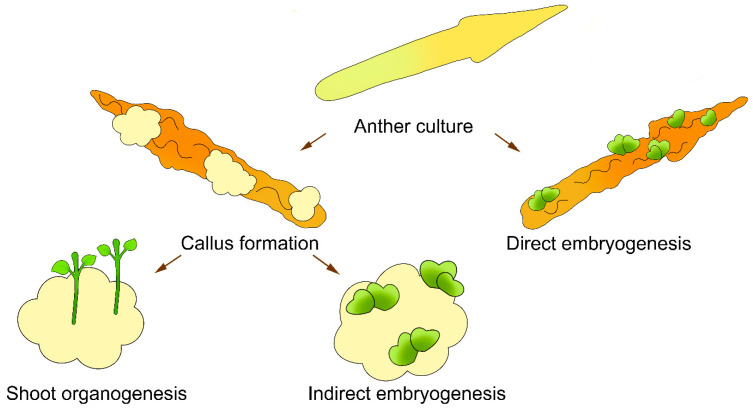
Morphogenic responses in sunflower anther culture.

**Figure 2 plants-11-02919-f002:**
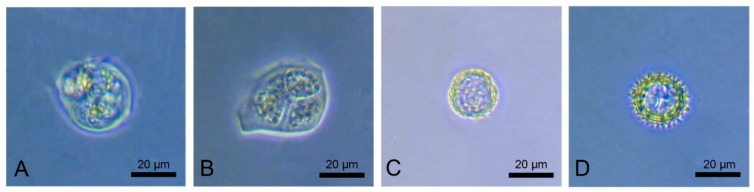
Stages of sunflower microspore development: (**A**) between meiotic dyad and tetrad; (**B**) tetrad of microspores; (**C**) early uninucleate microspore; and (**D**) late uninucleate microspore. Photographs were taken by authors of this review.

**Figure 3 plants-11-02919-f003:**
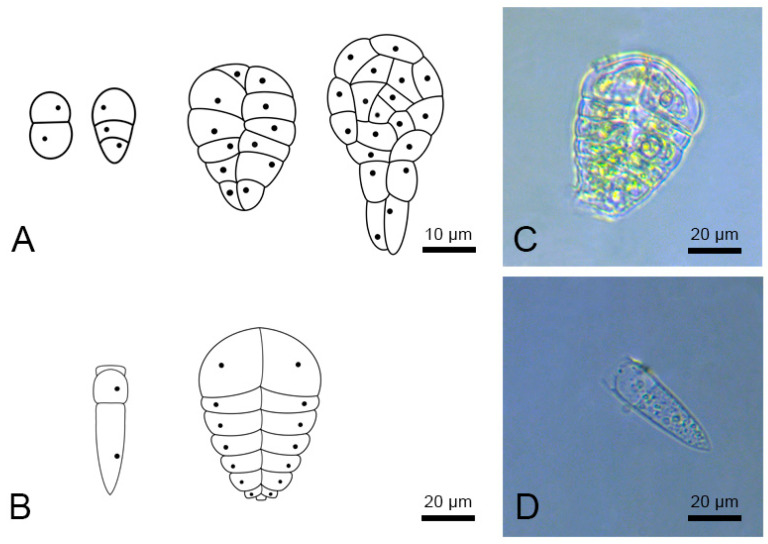
Similarity of gametic embryos and sunflower trichomes in vitro: (**A**) schematic of the developmental stages of a microspore-derived embryo; (**B**) schematic of the sunflower trichome morphology; (**C**,**D**) different morphologies of sunflower trichomes in a liquid medium. Photographs were taken by authors of this review.

**Figure 4 plants-11-02919-f004:**
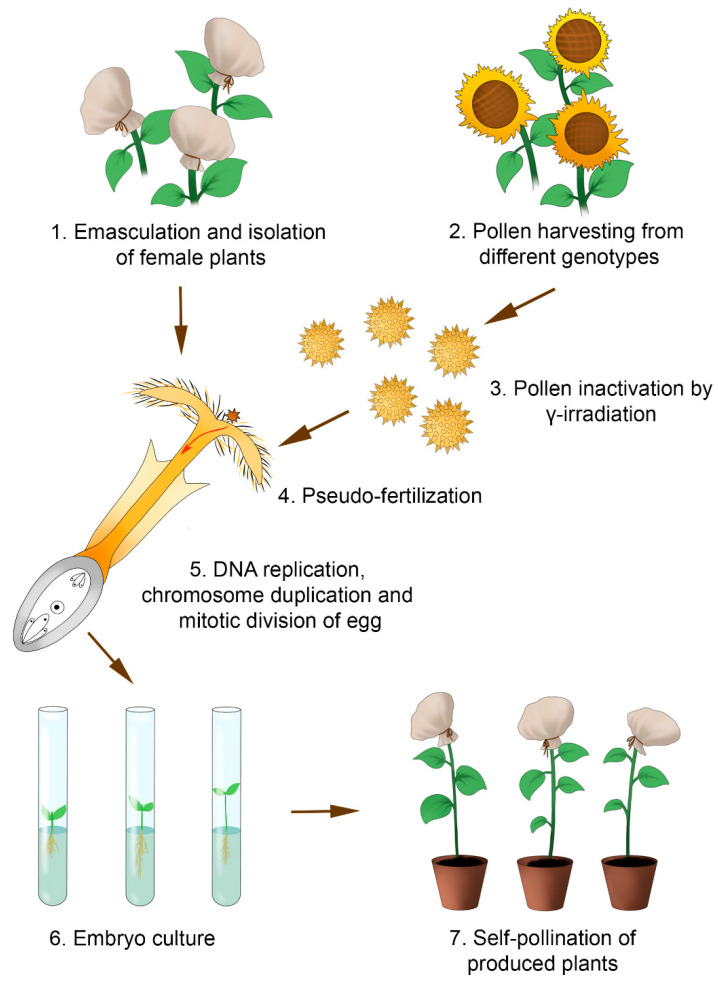
Stages of induced parthenogenesis of sunflower using γ-irradiated pollen.

**Figure 5 plants-11-02919-f005:**
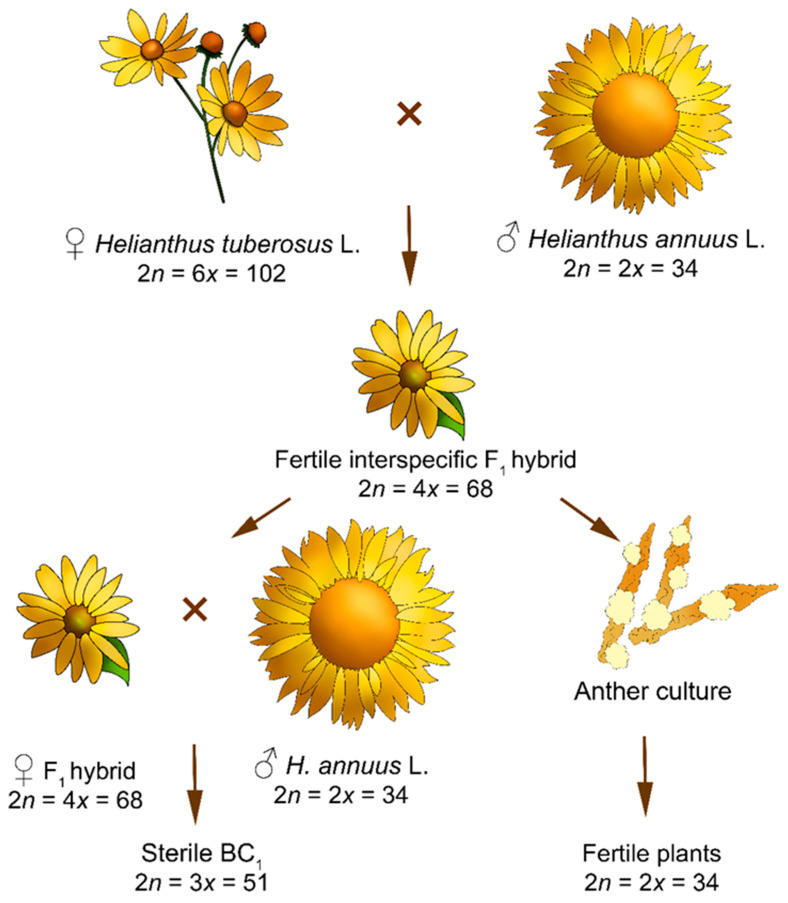
Manipulation of ploidy level in the anther culture of an interspecific hybrid between cultivated sunflower and *H. tuberosus* L. [17].

**Table 3 plants-11-02919-t003:** Brief characterization of protocols for producing gynogenic sunflower haploids.

Genotypes	Explants	Culture Conditions	Results	Reference
Eight sunflower cultivars	Unpollinated ovules and young florets	N6 + 0.125 mg/L MCPA + 2–6% sucrose	Ovules and young florets produced 96 and 12 embryos, respectively; produced plants were haploids and diploids	[75]
Eight sunflower cultivars	Unpollinated ovaries	For callus production and shoot regeneration: modified MS + 0.25 mg/L 2,4-D + 1 mg/L kinetin + 1 mg/L GA_3_ + 20 g/L sucrose; for direct embryogenesis: modified MS + 0.5 mg/L BAP + 30 g/L sucrose or modified MS + 1 mg/L IAA + 1 mg/L NAA + 0.5 mg/L kinetin + 1 mg/L BAP + 30 g/L sucrose	All genotypes formed calli (10–85%), 19 diploid plants were regenerated for three cultivars	[37]
Eight sunflower genotypes	Unpollinated ovaries	Modified MS + 2 mg/L NAA + 10% sucrose for ovaries; ovules were isolated from ovaries during cultivation and transferred to analogous medium	Embryogenesis frequency of 1.19%, 47% of embryos developed into plants: 50% haploid, 25% diploid, and 25% mixoploid	[76]
Not presented	Unpollinated ovules	Modified N6 + 12% sucrose for ovules; modified MS + 1 mg/L NAA + 1–2 mg/L BAP + 0.1 mg/L GA_3_ + 3–6% sucrose for formed embryos; modified MS + 1–2 mg/L BAP + 0.1 mg/L GA_3_ + 3–6% sucrose for shoot organogenesis	Up to 23.9% gynogenic embryos	[32]

**Table 4 plants-11-02919-t004:** Methods for determining the ploidy level and confirming the homozygosity of sunflower.

Method (Approach)	Description of Method	Results	References
**Direct methods for determining ploidy level**			
Chromosome count	Stain root tips or young leaves with acetocarmine or aceto-orcein, according to Feulgen [127], count chromosomes in the metaphase plates	Determination of haploids, diploids, aneuploids, and mixoploids	[9,17,39,42,47,48,76]
	Stain callus cells with acetocarmine, according to Feulgen	Detection of haploid and diploid cells, suggestive of spontaneous chromosome doubling	[14,42]
Flow cytometry	Isolate and stain nucleus, assess amount of DNA using a flow fluorometer	Data on ploidy level	[17,39,52,91,92,93,96]
**Indirect methods for determining ploidy level**			
Determination of morphological characteristics	Check phenotypic traits including height, anthocyanin coloration, leaf blade length and width, and leaf dentation	Differences between plants produced by anther culture and the donor plant	[9,38]
Histological analysis	Conduct histological examination of foci in preparations of cultivated anthers, ovules, and ovaries	Gametic embryo and callus from microspores and egg cells, somatic embryo and callus from anther walls, endothelium and integument tissue	[32,39,77]
Use of control plants	Use (1) emasculated plants not pollinated with irradiated pollen, or (2) plants pollinated with irradiated pollen but without embryo rescue	First control without seeds proves the effect of pseudo-pollination; second control with shrunken achene proves the absence of fertilization	[93,95]
**Methods for confirming plant homozygosity**			
Use of isozyme markers	Menadione reductase	Distinction between regenerants and donor plants	[38]
	Esterase and 6-phosphogluconate dehydrogenase	Confirmation of parthenogenetic origin	[91]
Use of molecular markers	Primers SSL26 and SSL46	Distinction between donor plants, paternal line, and doubled haploids	[94]

## Data Availability

Not applicable.

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
