# Peer review of "The Production of *Helianthus* Haploids: A Review of Its Current Status and Future Prospects"

_plants, 2022, doi:10.3390/plants11212919_

Round 1

Reviewer 1 Report

The review is devoted to obtaining doubled haploids in one of the most important oil crops in the world - Helianthus annuus L. Open sources provide scattered information on this direction in this crop. The authors have managed to systematize and analyze the currently available knowledge on obtaining haploid and doubled haploid Helianthus plants from male (anthers and isolated microspore culture in vitro) or female (unpollinated ovary and ovule culture in vitro) gametophytes and by induced parthenogenesis using γ-irradiated pollen and interspecific hybridization. Genetic, physiological and physical factors influencing the efficiency of producing haploid plants are also considered. The manuscript will be very useful for biotechnologists and plant breeders engaged in obtaining doubled haploids of sunflower.

There are some minor comments in the manuscript:

L 16 : should not be written "double haploids" but "doubled haploids" - correct throughout the manuscript

L 22 : instead of "isolated ovary and ovule cultures" better use the terminology "in vitro culture of unpollinated ovaries and ovules" - correct throughout the manuscript

L 139: Wrong "meso-inositol" should be written "Myo-inositol"- correct throughout the manuscript

L 367 "MS medium supplemented with 5 mL/L FeEDTA of the stock solution (5.57 g/L FeSO4 × 2H2O + 7.75 g/L Na2EDTA)" - the MS medium already contains NaFe-EDTA and it is not clear whether additional iron needs to be added to the culture medium? Also, the formula of the iron chelate is not correct, NaFe-EDTA should be written

Line 397: 16/8 photoperiod  16/8 h photoperiod

The male and female symbols are not clearly visible in the figure 5.

Line 768. It is not entirely clear what the authors mean by "broomrape races A–E". A, B, C etc. races?

Lines 803, 806. helianthi  Alternaria helianthi

Author Response

Dear Reviewer,

On behalf of ourselves and the co-authors, we thank you for your appreciation of our manuscript and for your valuable comments and questions. We are confident that your comments and corrections will make better our manuscript.

Remark 1. L 16 : should not be written "double haploids" but "doubled haploids" - correct throughout the manuscript

Response 1: Thank you for your valuable comments. We agree with the comments and corrected in the resubmitted text of the manuscript (L 16, 69, 78 etc.)

Remark 2. instead of "isolated ovary and ovule cultures" better use the terminology "in vitro culture of unpollinated ovaries and ovules" - correct throughout the manuscript.

Response 2: Thank you for your valuable comments. We agree with the comments and corrected in the resubmitted text of the manuscript. (L 23, 464, 474 etc.)

Remark 3. L 139: Wrong "meso-inositol" should be written "Myo-inositol"- correct throughout the manuscript. (L 342, Suppl. 1).

Response 3: Thank you for your valuable comments. We agree with the comments and corrected in the resubmitted text of the manuscript.

Remark 4. L 367 "MS medium supplemented with 5 mL/L FeEDTA of the stock solution (5.57 g/L FeSO4 × 2H2O + 7.75 g/L Na2EDTA)" - the MS medium already contains NaFe-EDTA and it is not clear whether additional iron needs to be added to the culture medium? Also, the formula of the iron chelate is not correct, NaFe-EDTA should be written

Response 4: Thank you for your valuable comments. We have tried to substantially simplify tables 1 and 2 for a better understanding for readers.

Remark 5. Line 397: 16/8 photoperiod → 16/8 h photoperiod

Response 5: Thank you for your valuable comments. We agree with the remark and have made corrections to the text (L 362, 369).

Remark 6. The male and female symbols are not clearly visible in the figure 5.

Response 6: Thank you for your valuable comments. We agree with the comments and have improved the quality of the fig. 5.

Remark 7. Line 768. It is not entirely clear what the authors mean by "broomrape races A–E". A, B, C etc. races?

Response 7: Thank you for your valuable comments. We have made appropriate corrections to the text (L 743).

Remark 8. Lines 803, 806. helianthi → Alternaria helianthin

Response 8: Thank you for your valuable comments. We have made appropriate corrections to the text (L 778, 781).

Once again, we are so grateful for your review and valuable comments. In addition, we also send a resubmitted Word document with your and other reviewers' comments.

Best regards,

Marat Khaliluev

Reviewer 2 Report

In this manuscript, the authors review the current status and future prospects of Helianthus haploid production. This manuscript could become an important contribution to our knowledge and help disseminate these very useful and effective techniques. Furthermore, there are very few recent reviews on it, so this work is particularly valuable. This document is well written, references are exhaustive and reliable, and I appreciate the vulgarization effort by schema of very good quality.

I have only a few minor comments which I believe are worth considering by authors in order to gain clarity and impact.

1-     A short more general paragraph at the beginning of the manuscript might be welcome to recontextualize the need for haploid production in plant/crop research with some references like:

-        Mishra, A. K., Saini, R., & Tiwari, K. N. (2021). Double Haploid Production and Its Applications in Crop Improvement. In Agricultural Biotechnology: Latest Research and Trends (pp. 75-101). Springer, Singapore.

-        Karjee, S., Mahapatra, S., Singh, D., Saha, K., & Viswakarma, P. K. (2020). Production of double haploids in ornamental crops. Journal of Pharmacognosy and Phytochemistry, 9(4), 555-565.

-        Hooghvorst, I., & Nogués, S. (2021). Chromosome doubling methods in doubled haploid and haploid inducer-mediated genome-editing systems in major crops. Plant Cell Reports, 40(2), 255-270.

-        Jacquier, N., Gilles, L. M., Martinant, J. P., Rogowsky, P. M., & Widiez, T. (2021). Maize In planta haploid inducer lines: A cornerstone for doubled haploid technology. Doubled Haploid Technology, 25-48.

2-     Table 1 and 2 footnotes are very difficult to read and understand, authors could use two different character types like 1,2,3,4… and a,b,c,… (not 1 vs1* ). I suggest authors to provide simpler tables and additional tables with all the information and notes that will be usefully to the community.

Author Response

Dear Reviewer,

On behalf of ourselves and the co-authors, we thank you for your appreciation of our manuscript and for your valuable comments. We are confident that your comments and corrections will make our manuscript better. We agree with the comments and questions indicated in the text of the manuscript. Below we present the corrections that were made to the resubmitted manuscript.

Remark 1: A short more general paragraph at the beginning of the manuscript might be welcome to recontextualize the need for haploid production in plant/crop research with some references like:…

Response 1: We thank the referee for the suggestion to complete the introduction section. However, in our opinion, we believe that this information will be redundant, since the review article is already large.

Remark 2. Table 1 and 2 footnotes are very difficult to read and understand, authors could use two different character types like 1,2,3,4… and a,b,c,… (not 1 vs1* ). I suggest authors to provide simpler tables and additional tables with all the information and notes that will be usefully to the community.

Response 2: We thank the referee for valuable remarks about tables 1 and 2. We have tried to substantially simplify tables 1 and 2 for a better understanding for readers. A detailed description of the genotypes, as well as the composition of culture media, including vitamins and amino acid mixture composition, were presented in the supplement 1 and 2, respectively.

Once again, we are so grateful for your review and valuable comments. In addition, we also send you a Word document with the text, taking into account of your comments and other Reviewers.

Best regards,

Marat Khaliluev
